# TEXT-TRANSPORT: Toward Learning Causal Effects of Natural Language

**Victoria Lin**
Carnegie Mellon University
vlin2@andrew.cmu.edu

**Louis-Philippe Morency**
Carnegie Mellon University
morency@cs.cmu.edu

**Eli Ben-Michael**
Carnegie Mellon University
ebenmichael@cmu.edu

## Abstract

As language technologies gain prominence in real-world settings, it is important to understand *how* changes to language affect reader perceptions. This can be formalized as the *causal effect* of varying a linguistic attribute (e.g., sentiment) on a reader's response to the text. In this paper, we introduce TEXT-TRANSPORT, a method for estimation of causal effects from natural language under any text distribution. Current approaches for valid causal effect estimation require strong assumptions about the data, meaning the data from which one *can* estimate valid causal effects often is not representative of the actual target domain of interest. To address this issue, we leverage the notion of distribution shift to describe an estimator that *transports* causal effects between domains, bypassing the need for strong assumptions in the target domain. We derive statistical guarantees on the uncertainty of this estimator, and we report empirical results and analyses that support the validity of TEXT-TRANSPORT across data settings. Finally, we use TEXT-TRANSPORT to study a realistic setting—hate speech on social media—in which causal effects do shift significantly between text domains, demonstrating the necessity of transport when conducting causal inference on natural language.

## 1 Introduction

What makes a comment on a social media site seem toxic or hateful (Mathew et al., 2021; Guest et al., 2021)? Could it be the use of profanity, or a lack of insight? What makes a product review more or less helpful to readers (Mudambi and Schuff, 2010; Pan and Zhang, 2011)? Is it the certainty of the review, or perhaps the presence of justifications? As language technologies are increasingly deployed in real-world settings, interpretability and explainability in natural language processing have become paramount (Rudin, 2019; Barredo Arrieta et al., 2020). Particularly desirable is the ability to

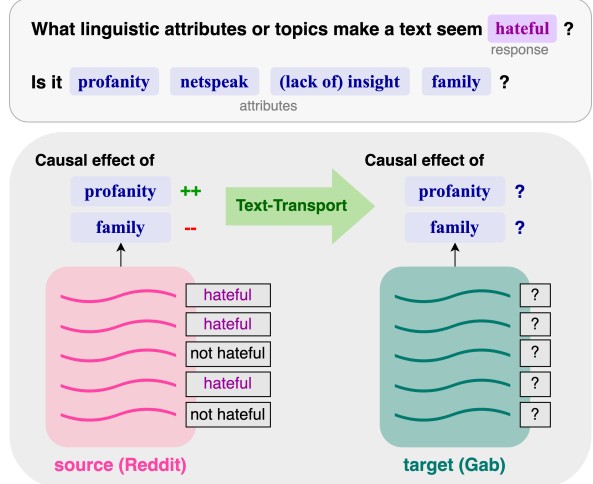

Figure 1: TEXT-TRANSPORT facilitates estimation of text causal effects on any target domain by *transporting* causal effects from a source domain.

understand *how* changes to language affect reader perceptions—formalized in statistical terms as the *causal effect* of varying a linguistic attribute on a reader's response to the text (Figure 1).

A core technical challenge for causal inference is that valid causal effects can only be estimated on data in which certain assumptions are upheld: namely, data where confounding is either fully measured or absent entirely (Rosenbaum and Rubin, 1983). Since confounding—the presence of factors that affect both the reader's choice of texts to read and the reader's response (e.g., political affiliation, age, mood)—is extremely difficult to measure fully in text settings, estimation of causal effects from natural language remains an open problem. One resource-intensive approach is to run a *randomized experiment*, which eliminates confounding by ensuring that respondents are randomly assigned texts to read (Holland, 1986; Fong and Grimmer, 2021). However, effects estimated from randomized experiments may not generalize outside of the specific data on which they were conducted (Tipton, 2014;

Bareinboim and Pearl, 2021). Therefore, learning causal effects for a new *target* domain can require a new randomized experiment each time.

In this paper, we propose to bypass the need for strong data assumptions in the target domain by framing causal effect estimation as a *distribution shift problem*. We introduce TEXT-TRANSPORT, a method for learning text causal effects in *any* target domain, including those that do not necessarily fulfill the assumptions required for valid causal inference. Leveraging the notion of distribution shift, we define a causal estimator that *transports* a causal effect from a causally valid source domain (e.g., a randomized experiment) to the target domain, and we derive statistical guarantees for our proposed estimator that show that it has desirable properties that can be used to quantify its uncertainty.

We evaluate TEXT-TRANSPORT empirically using a benchmarking strategy that includes knowledge about causal effects in both the source domain and the target domain. We find that across three data settings of increasing complexity, TEXT-TRANSPORT estimates are consistent with effects directly estimated on the target domain, suggesting successful transport of effects. We further study a realistic setting—user responses to hateful content on social media sites—in which causal effects do change significantly between text domains, demonstrating the utility of transport when estimating causal effects from natural language. Finally, we conduct analyses that examine the intuition behind why TEXT-TRANSPORT is an effective estimator of text causal effects.

## 2 Problem Setting

Consider a collection of texts (e.g., documents, sentences, utterances) $\mathcal{X}$, where person $i$ ($i = 1, \ldots, N$) is shown a text $X_i$ from $\mathcal{X}$. Using the potential outcomes framework (Neyman, 1923 [1990]; Rubin, 1974), let $Y_i(x)$ denote the potential *response* of respondent $i$ if they were to read text $x$, where $Y_i : \mathcal{X} \to \mathbb{R}$. Without loss of generality, assume that each respondent in reality reads only one text $X_i$, so their observed response is $Y_i(X_i)$.

Then the average response $\mu(P)$ across the $N$ respondents when texts $X$ are drawn from a distri-

bution $P$ is given by:

$$\mu(P) = \frac{1}{N} \sum_{i=1}^{N} E_{X \sim P}[Y_i(X)]$$

$$= \frac{1}{N} \sum_{i=1}^{N} \sum_{x \in \mathcal{X}} Y_i(x)P(x) \tag{1}$$

Now let $X$ be parameterized as $X = \{a(X), a^{\mathsf{c}}(X)\}$, where $a(X)$ is the text attribute of interest and $a^{\mathsf{c}}(X)$ denotes all other attributes of the text $X$. Note that for a text causal effect to be meaningful, $a(X)$ must be interpretable. This may be achieved by having a human code $a(X)$ or using a lexicon or other automatic coding method. Again for simplicity, we assume $a(X) \in \{0, 1\}$.

**Definition 1** (Natural causal effect of an attribute)**.** Let $P_1(X)$ be a distribution such that $a(X) = 1$ and $a^{\mathsf{c}}(X) \sim P(a^{\mathsf{c}}(X)|a(X) = 1)$, and let $P_0(X)$ be a distribution such that $a(X) = 0$ and $a^{\mathsf{c}}(X) \sim P(a^{\mathsf{c}}(X)|a(X) = 0)$.

Then the causal effect of $a(X)$ on $Y$ is given by:

$$\tau^* = \mu(P_1) - \mu(P_0) \tag{2}$$

Here, $\tau^*$ is the *natural effect* of $a(X)$. Linguistic attributes are subject to *aliasing* (Fong and Grimmer, 2021), in which some other linguistic attributes (e.g., the $k$-th linguistic attribute $a^{\mathsf{c}}(X)_k$) may be correlated with both the linguistic attribute of interest $a(X)$ and the response $Y$, such that $P(a^{\mathsf{c}}(X)_k|a(X) = 1) \neq P(a^{\mathsf{c}}(X)_k|a(X) = 0)$. For example, *optimism* may naturally co-occur with *positive emotion*, meaning that the natural effect of optimism also contains in part the effect of positive emotion. In contrast, the *isolated effect* would contain only the effect of optimism. In this paper, we choose to focus on natural effects due to the way linguistic attributes manifest in real-world language. That is, since optimism nearly always co-occurs with positive emotion, it may be difficult to interpret the effect of optimism when positive emotion is removed (the isolated effect), so we instead focus on their collective natural effect.

Our goal is then to learn $\tau_T$, the natural causal effect of the attribute $a(X)$ on response $Y$ in the text domain of interest $P^T$. We consider use cases where *it is not possible to directly estimate the effect from target data $X \sim P^T$*, either because $P^T$ does not fulfill the assumptions required for valid causal inference or simply because the response $Y$ is not measured in the domain of interest.

## 3 TEXT-TRANSPORT

To estimate causal effects under a target text distribution $P^T$—without computing effects on $P^T$ directly—we propose TEXT-TRANSPORT, a method for *transporting* a causal effect from a source text distribution $P^R$ that does fulfill the assumptions required for valid causal inference and with respect to which $P^T$ is absolutely continuous. Our approach can help to generalize the causal findings of $P^R$, which are specific to the data domain of $P^R$, to any text distribution of interest $P^T$. For mathematical convenience, we consider the source distribution $P^R$ to be a randomized experiment. We note that any *crowdsourced* dataset in which samples are randomly assigned to crowdworkers can be considered a randomized experiment.

### 3.1 Transporting effects

We characterize this problem as an instance of distribution shift, allowing us to define a causal effect estimator $\hat{\tau}^T$ that uses the *density ratio* between two distributions as an *importance weight* to transport the effect from $P^R$ to $P^T$. Given $X_i \sim P^R$, and letting $\frac{d\mathbb{P}^T}{d\mathbb{P}^R}(x) \equiv \frac{P^T(x)}{P^R(x)}$ be the density ratio[1] between $P^T$ and $P^R$,

$$\hat{\mu}(P^T) = \frac{1}{n}\sum_{i=1}^{n} \frac{d\mathbb{P}^T}{d\mathbb{P}^R}(X_i)Y_i(X_i) \qquad (3)$$

which gives us the effect estimate under $P^T$:

$$\hat{\tau}^T = \hat{\mu}(P_1^T) - \hat{\mu}(P_0^T) \qquad (4)$$

Intuitively, as all observed texts are drawn from $P^R$, the role of the importance weight is to increase the contribution of texts that are similar to texts from $P^T$, while reducing the contribution of texts that are representative of $P^R$. That is, if $P^T(X_i)$ is high and $P^R(X_i)$ is low, then $X_i$ will have a greater contribution to $\hat{\mu}(P^T)$. To highlight this transport of $P^R$ to $P^T$, in the remainder of this paper we will refer to $\hat{\mu}(P^T)$ as $\hat{\mu}^{R \to T}$.

A strength of this estimator is that we are able to quantify statistical uncertainty around the causal effect. We demonstrate (with derivations and proofs in Appendix B) that $\hat{\mu}^{R \to T}$ has a number of desirable properties that allow us to compute statistically valid confidence intervals: (1) it is an unbiased estimator of $\mu(P^T)$, (2) it is asymptotically normal, and (3) it has a closed-form variance and an unbiased, easy-to-compute variance estimator.

### 3.2 Importance weight estimation

Estimating the transported response $\hat{\mu}(P^T)$ first requires computing either the derivative $\frac{d\mathbb{P}^T}{d\mathbb{P}^R}(X)$ or the individual probabilities $P^R(X), P^T(X)$. While there are many potential ways to estimate this quantity, we propose the classification approach TEXT-TRANSPORT$_{\text{clf}}$ and the language model approach TEXT-TRANSPORT$_{\text{LM}}$ (Figure 2).

**TEXT-TRANSPORT$_{\text{clf}}$.** The classification approach for estimating $\frac{d\mathbb{P}^T}{d\mathbb{P}^R}(X)$ relies on the notion that the density ratio can be rewritten in a way that makes estimation more tractable. Let $C$ denote the distribution (or corpus) from which a text is drawn, where $C = T$ denotes that it is drawn from $P^T$ and $C = R$ denotes that it is drawn from $P^R$. Then $\frac{d\mathbb{P}^T}{d\mathbb{P}^R}(X)$ can be rewritten as follows:

$$\frac{d\mathbb{P}^T}{d\mathbb{P}^R}(X) = \frac{P(C = T|X)}{P(C = T)} \frac{P(C = R)}{P(C = R|X)} \qquad (5)$$

$P(C = T|X)$ and $P(C = R|X)$ can be estimated by training a binary classifier $M_\theta : \mathcal{X} \to \{0, 1\}$ to predict if a text $X$ came from $T$ or $R$. $P(C = R)$ and $P(C = T)$ are defined by their sample proportions (i.e., by their proportion of the total text after combining the two corpora).[2]

**TEXT-TRANSPORT$_{\text{LM}}$.** Because language models are capable of learning text distributions, we are able to take an alternative estimation approach that does not require learning $\frac{d\mathbb{P}^T}{d\mathbb{P}^R}(X)$. A language model trained on samples from $P^R$ or $P^T$, for instance, can compute the probability of texts under the learned distributions.

Pre-trained large language models (LLMs) are particularly useful for estimating $\hat{P}^R$ and $\hat{P}^T$, since their training corpora are large enough to approximate the distribution of the English language, and their training data is likely to have included many $P^R$ or $P^T$ of interest. Following recent advances in LLMs, one way of obtaining $\hat{P}^R(X)$ and $\hat{P}^T(X)$ from an LLM is to prompt the LLM in a way that induces it to focus on $P^R$ or $P^T$. Once the LLM has been prompted toward either $P^R$ or $P^T$, sentence probabilities from the LLM can be treated as reflections of $P^R(X)$ or $P^T(X)$, respectively. We provide examples of such prompts in Figure 2, and we explore this approach in our experiments.

---

[1]More generally, this is the *Radon-Nikodym derivative*, formally defined in Appendix A.

[2]In practice, rather than use the ratios $\frac{d\mathbb{P}^T}{d\mathbb{P}^R}(X)$ or $\frac{P^T(X_i)}{P^R(X_i)}$ directly, we use stabilized (i.e., normalized) versions that cancel out the sample proportions. See Appendix C.

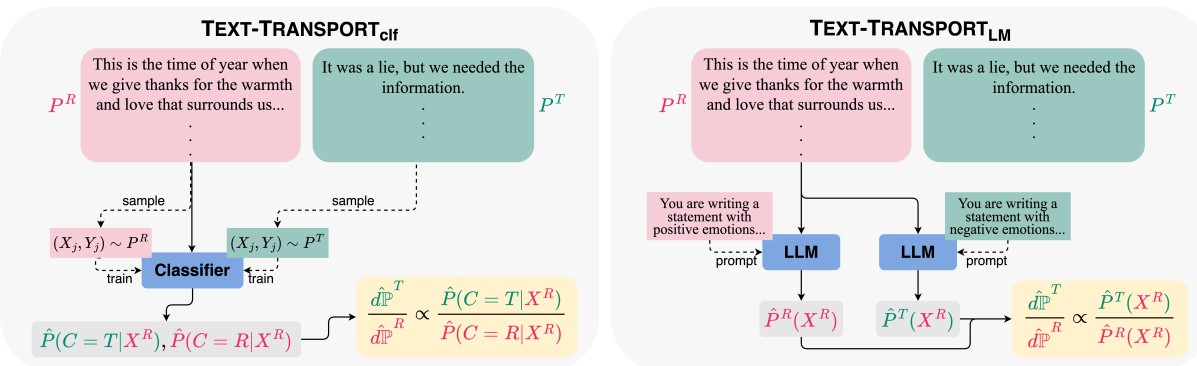

Figure 2: Two proposed approaches for estimating importance weights.

## 4 Experimental Setup

We conduct empirical evaluations to assess the validity of causal effects obtained through TEXT-TRANSPORT in three different data settings of increasing complexity (Table 1).

### 4.1 Evaluation methodology

To assess the validity of TEXT-TRANSPORT, we conduct experiments comparing the transported average response $\hat{\mu}^{R \to T}$ with $\hat{\mu}^R$, the average response under $P^R$, and $\hat{\mu}^T$, the average response under $P^T$. A valid transported response $\hat{\mu}^{R \to T}$ will be similar to $\hat{\mu}^T$ and significantly different from $\hat{\mu}^R$. To quantify these differences, we compare estimated averages and 95% confidence intervals, as well as normalized RMSE (RMSE divided by the standard deviation of the target response) between $\hat{\mu}^{R \to T}$ and $\hat{\mu}^T$.

As we mention previously, validating transported causal effects requires an evaluation strategy in which causal effects under $P^R$ and $P^T$ are both known. Therefore, each of our evaluation datasets consists of a single crowdsourced dataset that can be divided into two parts (e.g., a corpus of movie reviews can be split by genre), such that they possess the following properties. First, to allow $\hat{\mu}^R$ and $\hat{\mu}^T$ to be computed directly, both $P^R$ and $P^T$ are randomized experiments (we reiterate that $P^T$ is a randomized experiment only for the purposes of validation; we would not expect actual $P^T$s to be randomized). Second, the response $Y$ is measured for both $P^R$ and $P^T$. Third, $P^R$ and $P^T$ are distinct in a way where we would expect the *average* response to differ between the two. This allows us to evaluate whether transport has successfully occurred (if the average response is the same between $P^R$ and $P^T$, transport will not do anything).

| | $P^R$ | $P^T$ |
|---|---|---|
| Amazon | *Sound quality is likewise decent...* | *This printer being an all in one, serves several functions...* |
| EmoBank | *I have become more open-minded, more responsible...* | *I've even tried rewriting the corrupted sections...* |
| Hate Speech | *Your reply is a complete non-sequitur.* | *You really are a trained little monkey and don't even know it.* |

Table 1: Source ($P^R$) and target ($P^T$) distributions for each evaluation dataset.

We choose three crowdsourced datasets, partition each into $P^R$ and $P^T$, and compute $\hat{\mu}^R$, $\hat{\mu}^T$, and $\hat{\mu}^{R \to T}$. We estimate confidence intervals for each average response through bootstrap resampling with 100 iterations.

**Baselines.** While a small number of prior studies have proposed estimators of some type of text causal effect from observational (i.e., non-randomized) data, effects obtained from these methods are not directly comparable to those obtained using TEXT-TRANSPORT (further discussion of these methods can be found in Section 6). However, rather than using the density ratio to transport effects, other transformations of the source distribution to the target distribution are possible. One intuitive baseline is to train a predictive model on the source distribution, which is then used to generate pseudo-labels on the target distribution. These pseudo-labels can be averaged to produce a naive estimate (the **naive** baseline).

## 4.2 Datasets

**Amazon** (*controlled setting*). The Amazon dataset (McAuley and Leskovec, 2013) consists of text reviews from the Amazon e-commerce website, where the reader response is the number of "helpful" votes the review has received. We choose reviews of musical instruments as $P^R$ and reviews of office supplies as $P^T$.

To construct a best-case scenario in which there are no unmeasured factors in the data, we generate a new semi-synthetic response $Y$ by predicting the number of helpful votes as a function of $a(X), a^c(X)$. We use a noisy version of this prediction as our new $Y$. This ensures that all predictable variability in the response $Y$ is captured in the text. Furthermore, we sample reviews into $P^R$ and $P^T$ according to their predicted likelihood of being in $P^R$ or $P^T$ when accounting only for $a(X), a^c(X)$. This provides a controlled data setting in which we know that a model is capable of distinguishing between $P^R$ and $P^T$, such that we can evaluate TEXT-TRANSPORT under best-case conditions.

**EmoBank** (*partially controlled setting*). The EmoBank dataset (Buechel and Hahn, 2017) consists of sentences that have been rated by crowdworkers for their perceived *valence* $Y$ (i.e., the positivity or negativity of the text). To construct $P^R$, we sample sentences such that texts with high *writer-intended* valence occur with high probability, and to construct $P^T$, we sample sentences such that texts with low writer-intended valence occur with high probability. This partially controls the data setting by guaranteeing that the source and target domains differ on a single attribute—writer-intended valence—that is known to us (but hidden from the models).

**Hate Speech** (*natural setting*). The Hate Speech dataset (Qian et al., 2019) consists of comments from the social media sites Reddit and Gab. These comments have been annotated by crowdworkers as hate speech or not hate speech. The Reddit comments are chosen from subreddits where hate speech is more common, and Gab is a platform where users sometimes migrate after being blocked from other social media sites. To represent a realistic data setting in which we have no control over the construction of the source and target distributions, we treat the corpus of Reddit comments as $P^R$ and the corpus of Gab comments as $P^T$.

## 4.3 Implementation

**Linguistic attributes**. To automatically obtain linguistic attributes $a(X), a^c(X)$ from the text, we use the 2015 version of the lexicon Linguistic Inquiry and Word Count (**LIWC**) to encode the text as lexical categories (Pennebaker et al., 2015). LIWC is a human expert-constructed lexicon—generally viewed as a gold standard for lexicons—with a vocabulary of 6,548 words that belong to one or more of its 85 categories, most of which are related to psychology and social interaction. We binarize the category encodings to take the value 1 if the category is present in the text and 0 otherwise.

**TEXT-TRANSPORT_clf**. We use the following procedure to implement TEXT-TRANSPORT_clf. First, we consider data $\mathcal{D}_R$ from $P^R$ and data $\mathcal{D}_T$ from $P^T$. We take 10% of $\mathcal{D}_R$ and 10% of $\mathcal{D}_T$ as our classifier training set $\mathcal{D}_{train}$. Next, we train a classifier $M_\theta$ on $\mathcal{D}_{train}$ to distinguish between $P^R$ and $P^T$. For our classifier, we use embeddings from pre-trained MPNet (Song et al., 2020), a well-performing sentence transformer architecture, as inputs to a logistic regression.

From $M_\theta$, we can obtain $\hat{P}(C = T|X)$ and $\hat{P}(C = R|X)$ for all texts $X$ in the remaining 90% of $\mathcal{D}_R$. We compute $P(C = R)$ and $P(C = T)$ as $\frac{1}{|\mathcal{D}_R|}$ and $\frac{1}{|\mathcal{D}_T|}$, respectively. Then we have

$$\frac{d\mathbb{P}^R(X)}{d\mathbb{P}^T(X)} = \frac{\hat{P}(C = T|X)}{\hat{P}(C = R|X)} \frac{|\mathcal{D}_T|}{|\mathcal{D}_R|} \qquad (6)$$

In the case of the Amazon dataset, we note that although we can estimate the classification probabilities as $\hat{P}(C = T|X), \hat{P}(C = R|X)$, the true probabilities are already known, as we use them to separate texts into $P^R$ and $P^T$. Therefore—to evaluate the effectiveness of TEXT-TRANSPORT under conditions where we know the classifier to be correct—we use the known probabilities $P(C = T|X), P(C = R|X)$ in our evaluations on the Amazon dataset only.

**TEXT-TRANSPORT_LM**. This approach can be implemented without any training data, leaving the full body of text available for estimation. In our experiments, we estimate $P^R(X)$ and $P^T(X)$ through *prompting*. For each dataset, we provide pre-trained GPT-3 with a prompt that describes $P^R$ or $P^T$, then follow the prompt with the full text from each sample $X \sim P^R$. On the EmoBank dataset, for instance, we provide GPT-3 with the prompts "You are writing a positive statement" (for $P^R$, the high-valence distribution) and "You are

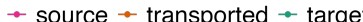

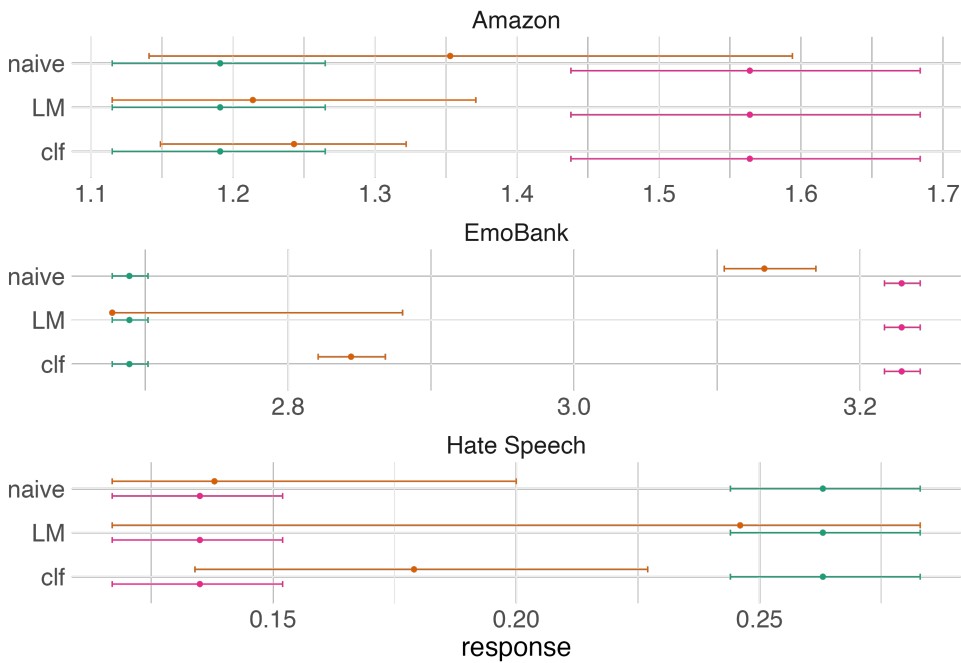

Figure 3: Validation of transported responses $\hat{\mu}^{R \to T}$ against known target responses $\hat{\mu}^T$ and source responses $\hat{\mu}^R$.

| | Amazon | EmoBank | Hate Speech |
|---|---|---|---|
| Naive baseline | 0.073 | 0.903 | 0.351 |
| TEXT-TRANSPORT$_{\text{clf}}$ | **0.019** | 0.832 | **0.257** |
| TEXT-TRANSPORT$_{\text{LM}}$ | 0.035 | **0.378** | 0.943 |

Table 2: Normalized RMSE of transported responses $\hat{\mu}^{R \to T}$ against known target responses $\hat{\mu}^T$.

writing a negative statement" (for $P^T$, the low-valence distribution). A full list of prompts can be found in Appendix D.3.

After prompting the model with either $P^R$ or $P^T$, we compute the token probabilities over each $X$, then compute sentence probabilities as the product of the token probabilities. If a text has multiple sentences, we treat the average of the sentence probabilities as the overall text probability. Finally, we compute the ratio $\frac{\hat{P}^T(X)}{\hat{P}^R(X)}$ as our importance weight.

## 5 Results and Discussion

### 5.1 Validity of TEXT-TRANSPORT

We evaluate the validity of our TEXT-TRANSPORT responses on the Amazon, EmoBank, and Hate Speech data settings (Figure 3, Table 2).

We observe that on the Amazon dataset, both TEXT-TRANSPORT$_{\text{clf}}$ and TEXT-TRANSPORT$_{\text{LM}}$ are well-validated. For both sets of Amazon re-

sults, our transported response $\hat{\mu}^{R \to T}$ is statistically significantly different from $\hat{\mu}^R$, while being statistically indistinguishable from $\hat{\mu}^T$. In contrast, the naive baseline produces an estimate with confidence intervals that overlap both $\hat{\mu}^R$ and $\hat{\mu}^T$, and its RMSE is higher than both TEXT-TRANSPORT estimates. The success of TEXT-TRANSPORT$_{\text{clf}}$ in this setting suggests that if the classifier is known to be a good estimator of the probabilities $P(C = T|X)$ and $P(C = R|X)$, the transported estimates will be correct. The success of TEXT-TRANSPORT$_{\text{LM}}$ in this setting, on the other hand, suggests that prompting GPT-3 can in fact be an effective way of estimating $P^R(X)$ and $P^T(X)$ and that the ratio between the two can also be used to produce valid transported estimates.

We further find that as the data setting becomes less controlled (i.e., EmoBank and Hate Speech), our transported responses continue to show encouraging trends—that is, the transported effect

indeed moves the responses away from $\hat{\mu}^R$ and toward $\hat{\mu}^T$, while transported responses from the naive baseline exhibit little to no movement toward the target. When evaluating TEXT-TRANSPORT$_{\text{LM}}$ on EmoBank, $\hat{\mu}^{R \to T}$ and $\hat{\mu}^T$ have no statistically significant difference. However, in evaluations of TEXT-TRANSPORT$_{\text{clf}}$, we find that $\hat{\mu}^{R \to T}$—though transported in the right direction—retains a statistically significant difference from $\hat{\mu}^T$; and when evaluating TEXT-TRANSPORT$_{\text{LM}}$ on Hate Speech, we observe wide confidence intervals for $\hat{\mu}^{R \to T}$ that cover both $\hat{\mu}^T$ and $\hat{\mu}^R$, though the point estimates of $\hat{\mu}^{R \to T}$ and $\hat{\mu}^T$ are very close.

Finally, we note that TEXT-TRANSPORT$_{\text{LM}}$ is less stable than TEXT-TRANSPORT$_{\text{clf}}$ with respect to the width of its confidence intervals, although the transported point estimates are better. This is particularly highlighted by the higher RMSE of TEXT-TRANSPORT$_{\text{LM}}$ compared to the naive baseline on the Hate Speech dataset, in spite of TEXT-TRANSPORT$_{\text{LM}}$'s much better point estimate. [3]

We posit that the instability of TEXT-TRANSPORT$_{\text{LM}}$ is due to the very small probability of any particular text occurring under a given probability distribution, as well as a potential lack of precision introduced when using prompting to target an LLM to a specific distribution. We observe that both $\hat{P}^R(X)$ and $\hat{P}^T(X)$ are typically both very small, and any difference between them—while minute in absolute terms—is amplified when taking their ratio. As a result, the range of importance weights $\frac{\hat{P}^T(X)}{\hat{P}^R(X)}$ under TEXT-TRANSPORT$_{\text{LM}}$ is much larger than the range of $\frac{d\hat{\mathbb{P}}^T}{d\hat{\mathbb{P}}^R}(X)$ under TEXT-TRANSPORT$_{\text{clf}}$, introducing a large amount of variability when estimating $\hat{\mu}^{R \to T}$.

Often, however, TEXT-TRANSPORT$_{\text{LM}}$ can still produce reasonable confidence intervals (as is the case for the Amazon and EmoBank datasets), and it illustrates the efficacy of the TEXT-TRANSPORT method under one of the simplest implementations (since no additional model training or even access to target data is required).

---

[3] In this sense, metrics like RMSE can be somewhat reductive, as they penalize the larger confidence interval but fail to capture the fact that the "transported" effect under the naive estimator has moved very little toward the target distribution, while the transported effect under the LM approach has correctly made a much larger shift toward the target distribution.

## 5.2 A realistic use case: What makes a comment hateful?

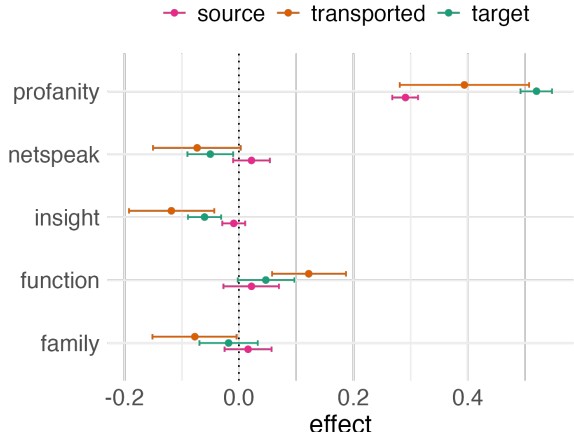

Figure 4: Source and transported natural causal effects from Reddit to Gab.

In this section, we highlight a realistic use case of TEXT-TRANSPORT. Taking our Hate Speech dataset, we examine the source, transported, and target effects of five linguistic attributes, where Reddit is again the source distribution and Gab is the target distribution (Figure 4). Transported effects are estimated using TEXT-TRANSPORT$_{\text{clf}}$.

We find that the causal effects of these five linguistic attributes estimated directly on Reddit differ significantly from their counterparts that have been transported to Gab. Though we would not have access to effects estimated directly on the target distribution in a real-world setting, we are able to validate here that the effect shifts are consistent with causal effects directly estimated from Gab.

*Negative causal effects.* After transport to Gab, the linguistic attributes *netspeak*, *insight*, and *family* are all shown to have significant negative effects on whether a comment is perceived as hate speech, while they are found to have no significant effect in the original Reddit data. In other words, when Gab users use netspeak, make insights, or talk about family, these conversational choices cause readers to perceive the overall comment as less hateful, while the same does not hold true for Reddit users.

*Positive causal effects.* In contrast, after transport to Gab, the linguistic attribute *function* is shown to have a significant positive effect on whether a comment is perceived as hate speech, while it was found to have no significant effect in the original Reddit data. Function words comprise articles, pronouns, conjunctions, negations,

| Classification approach | Language modeling approach |
|---|---|
| *They have the financial backing of powerful special interests and the determination to ignore the will of the people of Alaska and the American public.* | *Pushed as an effort to promote "Sonoma County" wines and a consumer education effort, the new law instead forces vintners to needlessly sully their package and undermine their own marketing efforts.* |
| *The efforts to save the mills failed partly because the international market for steel had largely dried up.* | *Yes, Self, I am also bothered that this observation ignores half-eaten cheese sandwiches, incomplete insect collections, and locks of infants' hair, forgotten in closets, basements, and warehouses.* |
| *Opera Cancelled Over a Depiction of Muhammad* | *Where is the authority?* |
| *7 Die As US Helicopter Crashes in Iraq* | *Suddenly, Amy screamed.* |

Table 3: Texts from the EmoBank source distribution $P^R$ (texts with high writer-intended valence) with the highest TEXT-TRANSPORT importance weights. Target distribution $P^T$ comprises texts with low writer-intended valence.

and other words that serve primarily grammatical purposes, and prior work has found that they can be highly suggestive of a person's psychological state (Groom and Pennebaker, 2002; Chung and Pennebaker, 2007; Pennebaker, 2011).

Though the difference between the original and transported effect is not statistically significant, *profanity* is also found to have a more positive effect on whether a comment is perceived as hate speech after transport to Gab compared to Reddit. This indicates that Gab users' use of profanity in their comments causes readers to perceive the overall comment as more hateful than if a Reddit user were to make profane remarks. This effect shift may be explained by the specific nature of the profanity used on Gab, which is qualitatively harsher and more offensive than the profanity used on Reddit.

The differences in these transported and original causal effects emphasize the importance of our method. An automatic content moderation algorithm, for instance, would likely need to consider different linguistic factors when deciding which comments to flag on each site.

### 5.3 An intuition for effect transport

Previously in Section 3.1, we stated that the intuition behind the change of measure $\frac{d\mathbb{P}^T}{d\mathbb{P}^R}(X)$ as an importance weight in the estimator $\hat{\mu}^{R \to T}$ was to increase the contribution of texts that are similar to $P^T$, while reducing the weight of texts that are most representative of $P^R$. To explore whether this is indeed the case, we identify the texts from EmoBank with the highest importance weights for each of our estimation approaches (Table 3). Texts with large importance weights have high $P^T(X)$

and low $P^R(X)$, meaning they should be similar to texts from $P^T$ despite actually coming from $P^R$.

We observe that texts from $P^R$ (i.e., texts with greater probability of high writer-intended valence) with high weights are in fact qualitatively similar to texts from $P^T$ (i.e., texts with greater probability of low writer-intended valence). That is, although texts from $P^R$ should be generally positive, the texts with the highest weights are markedly negative in tone, making them much more similar to texts from $P^T$. These observations support the intuition that the change of measure transports causal effects to the target domain by looking primarily at the responses to texts in the source domain that are most similar to texts in the target domain.

## 6 Related Work

TEXT-TRANSPORT draws on a diverse body of prior work from machine learning and causal inference, including the literature on domain adaptation, distribution shift, and generalizability and transportability. We build on methods from these fields to define a novel, flexible estimator of causal effects in natural language.

**Text causal effects**. A number of approaches have been proposed for estimating causal effects from natural language. Egami et al. (2018) construct a conceptual framework for causal inference with text that relies on specific data splitting strategies, while Pryzant et al. (2018) describe a procedure for learning words that are most predictive of a desired response. However, the interpretability of the learned effects from these works is limited. In a subsequent paper, Pryzant et al. (2021) introduce a method for estimating effects of linguistic proper-

ties from observational data. While this approach targets isolated effects of linguistic properties, it requires responses to be measured on the target domain, and it accounts only for the portion of confounding that is contained with the text.

Finally, in a recent paper, Fong and Grimmer (2021) conduct randomized experiments over text attributes to determine their effects. While allowing for valid causal inference, the resulting constructed texts are artificial in nature and constitute a clear use case for TEXT-TRANSPORT, which can transport effects from the less-realistic experimental text domain to more naturalistic target domains.

**Domain adaptation, distribution shift, and transportability**. Importance weighting has been widely used in the domain adaptation literature to help models learn under distribution shift (Byrd and Lipton, 2019). Models are trained with importance-weighted loss functions to account for covariate and label shift (Shimodaira, 2000; Lipton et al., 2018; Azizzadenesheli et al., 2019), correct for selection bias (Wang et al., 2016; Schnabel et al., 2016; Joachims et al., 2017), and facilitate off-policy reinforcement learning (Mahmood et al., 2014; Swaminathan and Joachims, 2015).

In parallel, a line of work studying the *external validity* of estimated causal effects has emerged within statistical causal inference (Egami and Hartman, 2022; Pearl and Bareinboim, 2022). These works aim to understand the conditions under which causal effects estimated on specific data can generalize or be transported to broader settings (Tipton, 2014; Bareinboim and Pearl, 2021). Prior work has also used density ratio-style importance weights to estimate average causal effects with high-dimensional interventions (de la Cuesta et al., 2022; Papadogeorgou et al., 2022).

We emphasize that TEXT-TRANSPORT is conceptually novel and methodologically distinct from these prior works. In our work, we explore an open problem—causal effect estimation from text—and define a new framework for learning causal effects from natural language. We propose a novel solution that uses tools from the domain adaptation and transportability literature, and we introduce novel methods for estimating natural effects in practice in language settings.

## 7  Conclusion

In this paper, we study the problem of causal effect estimation from text, which is challenging due to the highly confounded nature of natural language. To address this challenge, we propose TEXT-TRANSPORT, a novel method for estimating text causal effects from distributions that may not fulfill the assumptions required for valid causal inference. We conduct empirical evaluations that support the validity of causal effects estimated with TEXT-TRANSPORT, and we examine a realistic data setting—hate speech and toxicity on social media sites—in which TEXT-TRANSPORT identifies significant shifts in causal effects between text domains. Our results reinforce the need to account for distribution shift when estimating text-based causal effects and suggest that TEXT-TRANSPORT is a compelling approach for doing so. These promising initial findings open the door to future exploration of causal effects from complex unstructured data like language, images, and multimodal data.

## 8  Acknowledgements

This material is based upon work partially supported by the National Science Foundation (awards 1722822 and 1750439) and the National Institutes of Health (awards R01MH125740, R01MH132225, R01MH096951, and R21MH130767). Victoria Lin is partially supported by a Meta Research PhD Fellowship. Any opinions, findings, conclusions, or recommendations expressed in this material are those of the author(s) and do not necessarily reflect the views of the sponsors, and no official endorsement should be inferred.

## 9  Limitations

Although TEXT-TRANSPORT is effective in accounting for distribution shift to estimate effects of linguistic attributes in target domains without data assumptions, the method relies on the existence of a source domain that satisfies the data assumptions required for valid causal inference. Such a source domain—even a small or limited one—may not always be available. We plan to address this limitation in future work, where transport from any source domain is possible.

Additionally, TEXT-TRANSPORT proposes a framework for estimating *natural* causal effects from text. However, as we discuss above, in some cases it may also be of interest to estimate *isolated* causal effects from text. In future work, we will extend TEXT-TRANSPORT to include an estimator for isolated causal effects.

Finally, a requirement of the target distribution

$P^T$ is that it is absolutely continuous with respect to the source distribution $P^R$. The absolute continuity assumption is violated if a text that would *never* occur in $P^R$ could possibly occur in $P^T$. Therefore, this assumption may not be satisfied if the source and target distributions are completely unrelated and non-overlapping, even in latent space. Practically speaking, this means that it may not be possible to transport effects between distributions that are extremely different: for instance, from a corpus of technical manuals to a corpus of Shakespearean poetry.

## 10 Ethics Statement

**Broader impact**. Language technologies are assuming an increasingly prominent role in real-world settings, seeing use in applications like healthcare (Wen et al., 2019; Zhou et al., 2022; Reeves et al., 2021), content moderation (Pavlopoulos et al., 2017; Gorwa et al., 2020), and marketing (Kang et al., 2020). As these black-box systems become more widespread, interpretability and explainability in NLP are of ever greater importance.

TEXT-TRANSPORT builds toward this goal by providing a framework for estimating the causal effects of linguistic attributes on readers' responses to the text. These causal effects provide clear insight into *how changes to language affect the perceptions of readers*—an important factor when considering the texts that NLP systems consume or produce.

**Ethical considerations**. TEXT-TRANSPORT relies on pre-trained large language models to compute text probabilities. Consequently, it is possible that these text probabilities—which are used to transport causal effect estimates—may encode some of the biases contained in large pre-trained models and their training data. Interpretations of causal effects produced by TEXT-TRANSPORT should take these biases into consideration.

Additionally, we acknowledge the environmental impact of large language models, which are used in this work.

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

## A    Change of measure with Radon-Nikodym derivatives

The Radon-Nikodym derivative can be used to express one probability density function in terms of another probability density function, when the two densities are related by a change of measure. Specifically, if we have two probability measures defined on the same sample space, with one measure $\mathbb{P}$ absolutely continuous with respect to another measure $\mathbb{Q}$, then there exists a Radon-Nikodym derivative $Z$ such that:

$$\mathbb{P}(A) = \int_A Z d\mathbb{Q}$$

for any event $A$ in the sample space. Intuitively, this means that we can define the probability of any event under the measure $\mathbb{P}$ in terms of the probability of the same event under the measure $\mathbb{Q}$, by weighting the probabilities by a factor given by the Radon-Nikodym derivative.

Now, suppose we have two probability density functions $p(x)$ and $q(x)$ defined on some real-valued random variable $X$, with $q(x) > 0$ for all $x$. We want to express $p(x)$ in terms of $q(x)$ by a change of measure. To do this, we can define a new probability measure $\mathbb{P}$ as:

$$\mathbb{P}(A) = \int_A \frac{p(x)}{q(x)} q(x) dx = \int_A p(x) dx$$

for any event $A$ in the sample space. If $\mathbb{P}$ is absolutely continuous with respect to the measure defined by $q(x)$, then there exists a Radon-Nikodym derivative $Z(x)$ such that:

$$\frac{d\mathbb{P}}{d\mathbb{Q}}(x) = Z(x) = \frac{p(x)}{q(x)}$$

This means that we can express $p(x)$ in terms of $q(x)$ and the Radon-Nikodym derivative $Z(x)$ as:

$$p(x) = Z(x) q(x)$$

## B    Statistical properties of $\hat{\mu}^{R \to T}$

As we mention in the main paper, the transport estimator $\hat{\mu}^{R \to T}$ has a number of desirable statistical properties that allow us to quantify its uncertainty, included unbiasedness, asymptotic normality, and closed-form variance. In this section, we provide proofs and derivations for these properties.

### B.1    Unbiasedness

We can show that $\hat{\mu}^{R \to T}$ is an unbiased estimator for $\mu(P^T)$:

$$
\begin{aligned}
\mathbb{E}[\hat{\mu}^{R \to T}] &= \mathbb{E}_{X_i \sim P^R}[\hat{\mu}^{R \to T}] \\
&= \mathbb{E}_{X_i \sim P^R}\left[\frac{1}{n} \sum_{i=1}^{n} \frac{d\mathbb{P}^T}{d\mathbb{P}^R}(X_i) Y_i(X_i)\right] \\
&= \frac{1}{n} \sum_{i=1}^{n} \mathbb{E}_{X_i \sim P^R}\left[\frac{d\mathbb{P}^T}{d\mathbb{P}^R}(X_i) Y_i(X_i)\right] \\
&= \frac{1}{n} \sum_{i=1}^{n} \mathbb{E}_{X_i \sim P^T}[Y_i(X_i)] \\
&= \mu(P^T)
\end{aligned}
\tag{7}
$$

## B.2 Closed-form variance and confidence intervals

Let $\mathcal{X}$ be the space of all texts, and consider the finite-sample setting where $Y_i$ is fixed (i.e., non-random) for all $i \in [n]$. Then the variance of the estimator is given by:

$$
\begin{aligned}
\mathrm{Var}[\hat{\mu}^{R\to T}|Y_i] =&\, \mathrm{Var}\left[\frac{1}{n}\sum_{i=1}^{n}\frac{d\mathbb{P}^T}{d\mathbb{P}^R}(X_i)Y_i(X_i)\right] \\
=&\, \frac{1}{n^2}\sum_{i=1}^{n}\mathrm{Var}\left[\frac{d\mathbb{P}^T}{d\mathbb{P}^R}(X_i)Y_i(X_i)\right] \\
=&\, \frac{1}{n^2}\sum_{i=1}^{n}\mathrm{Cov}\left[\frac{d\mathbb{P}^T}{d\mathbb{P}^R}(X_i)Y_i(X_i),\frac{d\mathbb{P}^T}{d\mathbb{P}^R}(X_i)Y_i(X_i)\right] \\
=&\, \frac{1}{n^2}\sum_{i=1}^{n}\mathrm{Cov}\left[\sum_{x\in\mathcal{X}}\frac{d\mathbb{P}^T}{d\mathbb{P}^R}(x)Y_i(x)\mathbb{1}\{X_i=x\},\sum_{x'\in\mathcal{X}}\frac{d\mathbb{P}^T}{d\mathbb{P}^R}(x')Y_i(x')\mathbb{1}\{X_i=x'\}\right] \\
=&\, \frac{1}{n^2}\sum_{i=1}^{n}\sum_{x\in\mathcal{X}}\sum_{x'\in\mathcal{X}}\frac{d\mathbb{P}^T}{d\mathbb{P}^R}(x)\frac{d\mathbb{P}^T}{d\mathbb{P}^R}(x')Y_i(x)Y_i(x')\mathrm{Cov}[\mathbb{1}\{X_i=x\},\mathbb{1}\{X_i=x'\}] \\
=&\, \frac{1}{n^2}\sum_{i=1}^{n}\sum_{x\in\mathcal{X}}\sum_{x'\in\mathcal{X}}\frac{d\mathbb{P}^T}{d\mathbb{P}^R}(x)\frac{d\mathbb{P}^T}{d\mathbb{P}^R}(x')Y_i(x)Y_i(x')(\mathbb{E}[\mathbb{1}\{X_i=x\}\mathbb{1}\{X_i=x'\}] \\
&\, -\mathbb{E}[\mathbb{1}\{X_i=x\}]\mathbb{E}[\mathbb{1}\{X_i=x'\}]) \\
=&\, \frac{1}{n^2}\sum_{i=1}^{n}\sum_{x\in\mathcal{X}}\sum_{x'\in\mathcal{X}}\frac{d\mathbb{P}^T}{d\mathbb{P}^R}(x)\frac{d\mathbb{P}^T}{d\mathbb{P}^R}(x')Y_i(x)Y_i(x')(P^R(x,x')-P^R(x)P^R(x'))
\end{aligned}
$$

If $x=x'$,

$$
\begin{aligned}
\mathrm{Var}[\hat{\mu}^{R\to T}|Y_i] =&\, \frac{1}{n^2}\sum_{i=1}^{n}\sum_{x\in\mathcal{X}}\frac{d\mathbb{P}^{T2}}{d\mathbb{P}^R}(x)Y_i^2(x)(P^R(x)-P^R(x)P^R(x)) \\
=&\, \frac{1}{n^2}\sum_{i=1}^{n}\sum_{x\in\mathcal{X}}\frac{d\mathbb{P}^{T2}}{d\mathbb{P}^R}(x)Y_i^2(x)P^R(x)(1-P^R(x))
\end{aligned}
$$

If $x\neq x'$,

$$
\begin{aligned}
\mathrm{Var}[\hat{\mu}^{R\to T}|Y_i] =&\, \frac{1}{n^2}\sum_{i=1}^{n}\sum_{x\in\mathcal{X}}\sum_{\substack{x'\in\mathcal{X}\\x'\neq x}}\frac{d\mathbb{P}^T}{d\mathbb{P}^R}(x)\frac{d\mathbb{P}^T}{d\mathbb{P}^R}(x')Y_i(x)Y_i(x')(\underbrace{P^R(x,x')}_{0}-P^R(x)P^R(x')) \\
=&\, -\frac{1}{n^2}\sum_{i=1}^{n}\sum_{x\in\mathcal{X}}\sum_{\substack{x'\in\mathcal{X}\\x'\neq x}}\frac{d\mathbb{P}^T}{d\mathbb{P}^R}(x)\frac{d\mathbb{P}^T}{d\mathbb{P}^R}(x')Y_i(x)Y_i(x')P^R(x)P^R(x'))
\end{aligned}
$$

Putting the two cases together,

$$
\begin{aligned}
\text{Var}[\hat{\mu}^{R \to T}|Y_i] =& \frac{1}{n^2} \sum_{i=1}^{n} \left( \sum_{x \in \mathcal{X}} \frac{d\mathbb{P}^{T^2}}{d\mathbb{P}^R}(x) Y_i^2(x) P^R(x)(1 - P^R(x)) \right. \\
& \left. - \sum_{x \in \mathcal{X}} \sum_{\substack{x' \in \mathcal{X} \\ x' \neq x}} \frac{d\mathbb{P}^T}{d\mathbb{P}^R}(x) \frac{d\mathbb{P}^T}{d\mathbb{P}^R}(x') Y_i(x) Y_i(x') P^R(x) P^R(x') \right) \\
=& \frac{1}{n^2} \sum_{i=1}^{n} \left( \sum_{x \in \mathcal{X}} \frac{d\mathbb{P}^{T^2}}{d\mathbb{P}^R}(x) Y_i^2(x) P^R(x) - \sum_{x \in \mathcal{X}} \frac{d\mathbb{P}^{T^2}}{d\mathbb{P}^R}(x) Y_i^2(x) P^R(x)^2 \right. \\
& \left. - \sum_{x \in \mathcal{X}} \sum_{\substack{x' \in \mathcal{X} \\ x' \neq x}} \frac{d\mathbb{P}^T}{d\mathbb{P}^R}(x) \frac{d\mathbb{P}^T}{d\mathbb{P}^R}(x') Y_i(x) Y_i(x') P^R(x) P^R(x') \right) \\
=& \frac{1}{n^2} \sum_{i=1}^{n} \left( \sum_{x \in \mathcal{X}} \frac{d\mathbb{P}^{T^2}}{d\mathbb{P}^R}(x) Y_i^2(x) P^R(x) - \left( \underbrace{\sum_{x \in \mathcal{X}} \frac{d\mathbb{P}^{T^2}}{d\mathbb{P}^R}(x) Y_i^2(x) P^R(x)}_{\hat{\mu}_i = \hat{\mathbb{E}}_{x \sim PT}[Y_i(x)] = \hat{\mathbb{E}}_{x \sim PR}[\frac{dP^T}{dP^R}(x) Y_i(x)]} \right)^2 \right) \\
=& \frac{1}{n^2} \sum_{i=1}^{n} \left( \sum_{x \in \mathcal{X}} \left( \frac{d\mathbb{P}^T}{d\mathbb{P}^R}(x) Y_i(x) \right)^2 P^R(x) - \hat{\mu}_i^2 \right) \\
=& \frac{1}{n^2} \sum_{i=1}^{n} \left( \sum_{x \in \mathcal{X}} \left( \frac{d\mathbb{P}^T}{d\mathbb{P}^R}(x) Y_i(x) - \hat{\mu}_i \right)^2 P^R(x) + 2 \sum_{x \in \mathcal{X}} \frac{d\mathbb{P}^T}{d\mathbb{P}^R}(x) \hat{\mu}_i Y_i(x) P^R(x) \right. \\
& \left. - \sum_{x \in \mathcal{X}} \hat{\mu}_i^2 P^R(x) - \hat{\mu}_i^2 \right) \\
=& \frac{1}{n^2} \sum_{i=1}^{n} \left( \sum_{x \in \mathcal{X}} \left( \frac{d\mathbb{P}^T}{d\mathbb{P}^R}(x) Y_i(x) - \hat{\mu}_i \right)^2 P^R(x) + \hat{\mu}_i \left( 2 \underbrace{\sum_{x \in \mathcal{X}} \frac{d\mathbb{P}^T}{d\mathbb{P}^R}(x) Y_i(x) P^R(x)}_{\hat{\mu}_i} \right. \right. \\
& \left. \left. - \hat{\mu}_i \underbrace{\sum_{x \in \mathcal{X}} P^R(x)}_{1} - \hat{\mu}_i \right) \right) \\
=& \frac{1}{n^2} \sum_{i=1}^{n} \left( \sum_{x \in \mathcal{X}} \left( \frac{d\mathbb{P}^T}{d\mathbb{P}^R}(x) Y_i(x) - \hat{\mu}_i \right)^2 P^R(x) + \hat{\mu}_i(\underbrace{2\hat{\mu}_i - \hat{\mu}_i - \hat{\mu}_i}_{0}) \right) \\
=& \frac{1}{n^2} \sum_{i=1}^{n} \sum_{x \in \mathcal{X}} \left( \frac{d\mathbb{P}^T}{d\mathbb{P}^R}(x) Y_i(x) - \hat{\mu}_i \right)^2 P^R(x)
\end{aligned}
$$

Then finally, letting $\hat{\mu} = \hat{\mathbb{E}}_{x \sim PT}\left[ \frac{1}{n} \sum_{i=1}^{n} Y_i(x) \right] = \hat{\mathbb{E}}_{x \sim PR}\left[ \frac{1}{n} \sum_{i=1}^{n} \frac{dP^T}{dP^R}(x) Y_i(x) \right]$, we have

$$
\begin{aligned}
\text{Var}[\hat{\mu}^{R \to T}] &= \mathbb{E}_Y[\text{Var}[\hat{\mu}(P)|Y_i]] \\
&= \mathbb{E}_Y \left[ \frac{1}{n^2} \sum_{i=1}^{n} \sum_{x \in \mathcal{X}} \left( \frac{d\mathbb{P}^T}{d\mathbb{P}^R}(x) Y_i(x) - \hat{\mu}_i \right)^2 P^R(x) \right] \\
&= \frac{1}{n^2} \sum_{i=1}^{n} \sum_{x \in \mathcal{X}} \mathbb{E}_Y \left[ \left( \frac{d\mathbb{P}^T}{d\mathbb{P}^R}(x) Y_i(x) - \hat{\mu} \right)^2 \right] P^R(x)
\end{aligned}
\tag{8}
$$

With the central limit theorem (CLT), we establish asymptotic normality:

$$\frac{\hat{\mu}^{R \to T} - \mu(P^T)}{\sqrt{\mathrm{Var}[\hat{\mu}^{R \to T}]}} \to N(0, 1) \tag{9}$$

which we can use to estimate confidence intervals using the following unbiased variance estimate:

$$\widehat{\mathrm{Var}}[\hat{\mu}^{R \to T}] = \frac{1}{n^2} \sum_{i \in [n]} \left( \frac{d\hat{\mathbb{P}}^T}{d\mathbb{P}^R}(X_i)Y_i(X_i) - \hat{\mu}_i \right)^2 \tag{10}$$

## C   Hajek estimators

In practice, to maintain the stability of the importance weights (which can be very small), the Hájek (1971) estimator is often used in place of the instead of the standard Horvitz-Thompson estimator. With the Hajek estimator, the importance weights are normalized by the *average* importance weight. Then letting the importance weight be denoted by $\gamma$, we have the estimator

$$\hat{\mu}^{R \to T} = \frac{1}{n} \sum_{i=1}^{n} \gamma_i(X_i)Y_i(X_i)$$

For TEXT-TRANSPORT$_{\mathrm{clf}}$,

$$\hat{\gamma}_i = \frac{d\hat{\mathbb{P}}^T}{d\mathbb{P}^R}(X_i) \bigg/ \left( \frac{1}{n} \sum_{j=1}^{n} \frac{d\hat{\mathbb{P}}^T}{d\mathbb{P}^R}(X_j) \right)$$

and for TEXT-TRANSPORT$_{\mathrm{LM}}$,

$$\hat{\gamma}_i = \frac{\hat{P}^T(X_i)}{\hat{P}^R(X_i)} \bigg/ \left( \frac{1}{n} \sum_{j=1}^{n} \frac{\hat{P}^T(X_j)}{\hat{P}^R(X_j)} \right)$$

## D   Experimental Details

### D.1   Data

|  | $\mathcal{D}_R$ | $\mathcal{D}_T$ | $\mathcal{D}_{train}$ | $n_{total}$ | License |
|---|---|---|---|---|---|
| Amazon | 2,561 | 5,000 | 889 | 7,561 | Unknown |
| EmoBank | 3,350 | 1,320 | 529 | 4,670 | CC-BY-SA 4.0 |
| Hate Speech | 22,250 | 33,776 | 5,415 | 56,026 | CC BY-NC 4.0 |

Table 4: Composition of data splits. For each dataset, the number of samples in $D_R$, $D_T$, and $D_{train}$ is given, along with total samples for each dataset. Licensing information is also provided.

Details of our datasets are provided in Table 4, including dataset composition and licensing information. All three datasets are publicly available, and all are in English.

### D.2   Model details

**TEXT-TRANSPORT$_{\mathrm{clf}}$.**     We   used   HuggingFace's   implementation   of   MPNet   in   its `sentence-transformers` library (version 2.2.2), using the pre-trained model `all-mpnet-base-v2`. Embeddings from MPNet are 768 dimensions. Our logistic regression classifier was implemented in `scikit-learn` (version 1.0.2). All hyperparameters were set to their default values.

**TEXT-TRANSPORT$_{\mathrm{LM}}$.** We used `text-davinci-003` from the OpenAI API (version 0.27.4) as our pre-trained GPT-3. We prompted GPT-3 and computed sentence probabilities through the API. We set temperature to 0 and the maximum number of generated tokens to 0, since we wanted the model to echo our text input rather than generate new texts.

## D.3 Prompts

|  | $P^R$ prompt | $P^T$ prompt |
|---|---|---|
| Amazon | *You are writing a review for your purchase of a musical instrument on Amazon. Consider the following sentence.* | *You are writing a review for your purchase of an office product on Amazon. Consider the following sentence.* |
| EmoBank | *You are writing a positive statement. Consider the following sentence.* | *You are writing a negative statement. Consider the following sentence.* |
| Hate Speech | *You are writing a comment on a toxic subreddit of the social media site Reddit. Consider the following sentence.* | *You are writing a comment on the alt-right social media site Gab. Consider the following sentence.* |

Table 5: The full list of prompts provided to GPT-3 to induce them to focus on $P^R$ and $P^T$ for each of the evaluation datasets.

|  | 25% Quantile | Median | 75% Quantile |
|---|---|---|---|
| Amazon | 8.429 | 997.189 | $3.013 \times 10^5$ |
| EmoBank | 0.223 | 8.780 | 225.346 |
| Hate Speech | 0.120 | 1.515 | 23.268 |

Table 6: Probability ratios between GPT-3 language models that have been given a $P^R$ prompt and models that have been given a $P^T$ prompt. A median ratio greater than 1 suggests that prompting has been successful.

The prompts we used to induce GPT-3 toward the source distribution $P^R$ and the target distribution $P^T$ are provided in Table 5. To confirm that these prompts indeed induce GPT-3 to move toward $P^R$ or $P^T$, we conducted the following empirical validation. Given text $X^R \sim P^R$, we computed the ratio between $P(X^R)$ for GPT-3 that had been given a $P^R$ prompt and $P(X^R)$ for GPT-3 that had been given a $P^T$ prompt—in other words, the probability ratio $\frac{P_{\text{GPT-3}_{P^R}}(X^R)}{P_{\text{GPT-3}_{P^T}}(X^R)}$.

Since the texts $X^R$ are drawn from $P^R$, then if the prompts indeed direct GPT-3 toward the intended distribution, we would expect this ratio to have a median value greater than 1, as $P^R(X^R)$ should be larger than $P^T(X^R)$. We report medians and quantiles across the three evaluation datasets in Table 6.

We observe that across all three datasets, the median ratio is in fact greater than 1, indicating that our prompting strategy is successfully targeting GPT-3 to $P^R$ or $P^T$. The median ratio for the Hate Speech dataset—while still greater than 1—is much closer to 1 than the Amazon or EmoBank datasets, which is consistent with the intuition that targeting very specific distributions like Reddit and Gab with prompting can be more challenging.

## D.4 Computing resources

All experiments were conducted on machines with consumer-level NVIDIA graphics cards. We estimate the number of GPU hours used in our experiments to be fewer than 10.