# OpenReview forum: "Text-Transport: Toward Learning Causal Effects of Natural Language"
_EMNLP/2023/Conference — EMNLP 2023 Main_

### Official Review · Reviewer_g3EB · 2023-08-05

**Soundness:** 4

**Excitement:**

4: Strong: This paper deepens the understanding of some phenomenon or lowers the barriers to an existing research direction.

**Paper Topic And Main Contributions:**

The paper proposes a method to estimate causal effects of linguistic attributes in a target text distribution given measured causal effects from a source text distribution (Text-Transport): reweight observations from the source distribution more highly when similar to the target texts and vice versa (density ratio). (Derivations for the causal estimator are provided in the appendix.) To estimate this, two methods are proposed: classification (clf), based on a binary classifier that identifies if a text came from source or target, and LM, based on prompting a LLM and using the output probabilities. Through experiments on a range of controlled (Amazon reviews, tweet valence) and natural (hate speech) settings, the results (a) show that their method for transporting causal effects in controlled settings generally align with the actual effects; (b) discuss LIWC attributes with causal impact for hate speech detection on Reddit vs. Gab.

**Questions For The Authors:**

* For the LLM-based method, did you assess if prompted output is indeed more similar to texts from the target vs. source corpora either qualitatively or quantitatively?
* Not a question, but please feel free to engage with other points above and/or note if I've misunderstood/misrepresented anything from your paper in this review.

**Reasons To Accept:**

* The method for transporting causal effects and derivation seem to be sound and seem, as far as I am aware, to be novel. (Note that I have some familiarity but am not an expert on causal methods.)

* The experimental results generally seem to support the claims made in the paper (though see comments on the LLM-based estimator below).

* The paper is clearly written, organized well, and situated in existing literature on measuring causal effects in text as well as domain adaptation / distribution shift.

**Reasons To Reject:**

The intuition behind a classification-based estimator makes sense to me; I'm a bit more hesitant about the LLM-based one, both in principle and results. In principle, the paper states that LLM training corpora are "large enough to approximate the distribution of the English language" (line 230), and for certain domains (like hate speech) I'm not sure that's the case, given possible filtering of training data etc. It's not obvious that prompting to generate hate speech on Reddit or Gab really generates distributions close to a per-platform basis vs. just hate speech more generally, and the results on hate speech seem to support that (the confidence interval for the LLM-based estimator spans both domains).

_edit, after author response_: the response (and answer to my question below) has helped clarify both the purpose of the LLM-based estimator and its performance on the hate speech data.

**Reproducibility:**

4: Could mostly reproduce the results, but there may be some variation because of sample variance or minor variations in their interpretation of the protocol or method.

**Reviewer Confidence:**

2: Willing to defend my evaluation, but it is fairly likely that I missed some details, didn't understand some central points, or can't be sure about the novelty of the work.

---

> ### Author Rebuttal · Authors · 2023-08-28
>
> Thank you for your feedback and positive comments!
>
> [**LLM prompting**] Thank you for this point! We chose to use Text-Transport with an LLM prompting-based approach to illustrate the efficacy of the method with one of the "simplest" implementations (since no additional model training or even access to target data is required). It is true that using prompting to target an LLM to a specific distribution may not always be precise—as may be the case for the Hate Speech dataset, where the transported point estimate is very close to the target point estimate (suggesting some targeting signal) but the confidence intervals are wide (suggesting lack of precision). However, we believe that the success of the LLM approach in two additional data settings demonstrates that there is merit to the approach, and we see the question of how LLM targeting (or the use of LLMs in this context more generally) can be refined as an opportunity for future work.
>
> [**Question 1, assessing prompted output**] Qualitatively, we did observe that LLMs that had been prompted toward the EmoBank target distribution (low valence) found texts with low valence to be more probable (see Table 2 of our paper). To support these observations, we report additional quantitative evidence below. Let $X^R$ denote a text drawn from $P^R$. We report the ratio of $P(X^R)$ for an LLM that has been prompted toward $P^R$ versus an LLM that has been prompted toward $P^T$—in other words, the quantity $\frac{P\_{LLM\_{P^R}}(X^R)}{P\_{LLM\_{P^T}}(X^R)}$.
>
> If prompting indeed directs the LLM toward the intended distribution, we would expect this ratio to have a median value greater than 1. We report medians and quantiles across the three evaluation datasets:
>
> | Dataset 	| 25% Quantile |  Median | 75% Quantile |
> |-------------|:------------:|:-------:|:------------:|
> | Amazon  	| 	8.429	| 997.189 | 3.013*10^5   |
> | EmoBank 	| 	0.223	|  8.780  | 225.346  	|
> | Hate Speech | 	0.120	|  1.515  | 23.268   	|
>
> We observe that across all three datasets, the median ratio is greater than 1; and for the Amazon and EmoBank datasets, the ratio is much greater than 1. The median ratio for the Hate Speech dataset—while still greater than 1—is indeed closer to 1 than the others, which is consistent with the reviewer’s intuition that targeting very specific distributions like Reddit and Gab with prompting can be more challenging.

---

### Official Review · Reviewer_aAKu · 2023-08-07

**Soundness:** 4

**Excitement:**

4: Strong: This paper deepens the understanding of some phenomenon or lowers the barriers to an existing research direction.

**Paper Topic And Main Contributions:**

This paper tackles the problem of domain shift under causal estimation from text. The proposed framework, TEXT-TRANSPORT, calculates the density ratio between source and target domains via a trained classifier or a prompted LLM to distinguish where the given text comes from. The authors validate their framework empirically on three datasets and further demonstrate a use case of their proposed framework on hate speech detection.

**Questions For The Authors:**

1. The probability produced by LLM is known to be poorly calibrated. How do you think will it affect the validity of your approach? Any possible workarounds?

2. How would you extend your framework to a more complicated setting where you have m known domains and n unknown domains to be transferred? (In your paper you assume that m=n=1)

**Reasons To Accept:**

1. A novel framework to facilitate causal effect estimation from text under different domains, which could be potentially useful for computational social science and causal NLP research community.

2. Empirical results on three datasets validate the effectiveness of their proposed method.

**Reasons To Reject:**

1. Lack of comparison to some naive estimator baselines to further validate the effectiveness of the proposed framework. The paper only shows results on an estimator derived from eq 3. A direct baseline would be: applying a trained classifier (e.g. helpfulness prediction classifier for Amazon dataset) on the source domain to produce pseudo labels on the target domain, then using the pseudo labels to conduct randomized experiments to compute the causal effect.

2. Evaluation metrics of the validity could be more comprehensive. Besides confidence interval, I imagine that computing instance-level estimation (ITE) error could also help to show the accuracy of the transport estimator.

**Reproducibility:**

4: Could mostly reproduce the results, but there may be some variation because of sample variance or minor variations in their interpretation of the protocol or method.

**Reviewer Confidence:**

3: Pretty sure, but there's a chance I missed something. Although I have a good feel for this area in general, I did not carefully check the paper's details, e.g., the math, experimental design, or novelty.

---

> ### Author Rebuttal · Authors · 2023-08-28
>
> Thank you for your review! We appreciate your positive feedback and constructive questions.
>
> [**Comparison to naive estimator**] Thank you for this suggestion! We report the results of this naive baseline below, in which a predictive model trained on the source distribution is used to produce pseudo-labels on the target distribution, which are then averaged. We will include these results in the next version of our paper. Following the reviewer’s suggestion of additional evaluation metrics, we also report the normalized RMSE (RMSE divided by the standard deviation of the target response) of our transported response $\hat{\mu}_{R \rightarrow T}$ compared to the known target $\hat{\mu}_T$ across 100 bootstrap iterations.
>
> | Dataset 	| Method | Distribution | Response | Lower CI | Upper CI |  RMSE |
> |-------------|:------:|:------------:|:--------:|:--------:|:--------:|:-----:|
> | Amazon  	|	-   |	source	| 1.564	| 1.438	| 1.684	|   -   |
> | Amazon  	|	-   |	target	| 1.191	| 1.115	| 1.265	|   -   |
> | Amazon  	|   clf  |  transported | 1.243	| 1.149	| 1.322	| 0.019 |
> | Amazon  	|   LM   |  transported | 1.214	| 1.066	| 1.371	| 0.035 |
> | Amazon  	|  naive |  transported | 1.353	| 1.141	| 1.594	| 0.073 |
> | EmoBank 	|	-   |	source	| 3.229	| 3.217	| 3.242	| - 	|
> | EmoBank 	|	-   |	target	| 2.689	| 2.677	| 2.702	| - 	|
> | EmoBank 	|   clf  |  transported | 2.844	| 2.821	| 2.868	| 0.832 |
> | EmoBank 	|   LM   |  transported | 2.677	| 2.470	| 2.88 	| 0.378 |
> | EmoBank 	|  naive |  transported | 3.135	| 3.11 	| 3.161	| 0.903 |
> | Hate Speech |	-   |	source	| 0.135	| 0.117	| 0.152	| - 	|
> | Hate Speech |	-   |	target	| 0.263	| 0.244	| 0.283	| - 	|
> | Hate Speech |   clf  |  transported | 0.179	| 0.134	| 0.227	| 0.257 |
> | Hate Speech |   LM   |  transported | 0.246	| 0.000	| 1.000	| 0.943 |
> | Hate Speech |  naive |  transported | 0.138| 0.080	| 0.200	| 0.351 |
>
> We can see that in contrast to our method, Text-Transport, the naive estimator produces effect estimates that are farther from the true target effects. In fact, for the EmoBank and Hate Speech datasets, there is essentially no movement of the transported estimate toward the target distribution, while for the Amazon dataset, there is less movement compared to either implementation of the Text-Transport approach. Interestingly, the RMSE highlights one particular result: the lower RMSE of the naive estimator on the Hate Speech dataset compared to the Text-Transport LM approach, which is due to the larger confidence interval of the LM approach. This type of metric penalizes the larger confidence interval but fails to capture the fact that the "transported" effect under the naive estimator has moved very little toward the target distribution, while the transported effect under the LM approach has correctly made a much larger shift toward the target distribution. In this sense, metrics like RMSE can be somewhat reductive.
>
> Additionally, we note that in future work, the type of predictive modeling used in the naive estimator can be incorporated into the Text-Transport method via the *doubly robust estimator*, defined as follows:
>
> $$\hat{\mu}(P^T)_{DR}=\frac{1}{n}\sum\_{i=1}^n \frac{\hat{d\mathbb{P}^T}}{d\mathbb{P}^R}(X_i)(Y_i(X_i)-\hat{m}(X_i)) +\frac{1}{n}\sum\_{i=1}^n\hat{m}(X_i)$$
>
> The doubly robust estimator is commonly used in statistical causal inference to reduce the variance of estimates. By using a learned outcome model $\hat{m}(X)$, the doubly robust estimator ensures that the variance of the estimate is a function of the difference between the ground truth outcome $Y(X)$ and the predicted outcome $\hat{m}(X)$, rather than a function of $Y(X)$ itself.
>
> [**Instance-level estimation error**] Under this framework, instance-level estimation error cannot be computed because there is no 1:1 mapping between points in the source distribution and points in the target distribution. However, we hope that our inclusion of RMSE above—and the results of the naive estimator—will provide further evidence for the validity of Text-Transport.
>
> [**Question 1**] While some prior research has suggested that probabilities from LLMs are poorly calibrated, our empirical results indicate that these probabilities still contain sufficient signal to successfully transport effect estimates across text distributions. Improving the probability calibration of LLMs is also an active area of research, and very recently, works have shown that the newest LLMs (e.g., GPT-4 in Nori et al. 2023) display better probability calibration compared to previous LLMs. We note that the implementation of the LLM approach described in this paper is the first step in a longer research agenda on estimating text causal effects, and ongoing improvements in LLM probability calibration also pose interesting future directions for our work.
>
> Harsha Nori, Nicholas King, Scott Mayer McKinney, Dean Carignan & Eric Horvitz. (2023) Capabilities of GPT-4 on Medical Challenge Problems.
>
> [**Question 2**] Introducing additional source and target domains could be as straightforward as training a multi-class classifier over multiple text domains (for the classification approach) or prompting the LLM to target multiple text domains. We chose $m=n=1$ for evaluation purposes only; the Text-Transport approach is not inherently limited to a single source and target domain. The $m > 1$ case does offer several interesting questions about aggregating across estimates. For example, if $m >1$ and $n = 1$, directly using Text-Transport would give $m$ different estimates of the effect in the target domain. While it is valid to simply average these different estimates together, alternatives such as precision weighting may perform better. Future work can help adjudicate what methods of aggregation work best.

---

### Official Review · Reviewer_VqXU · 2023-08-12

**Soundness:** 4

**Excitement:**

4: Strong: This paper deepens the understanding of some phenomenon or lowers the barriers to an existing research direction.

**Paper Topic And Main Contributions:**

This paper is concerned with determining the causal effects of language properties on the response of the reader. For example, how the presence of profanity may affect the reader's perception of the text as hate speech. Estimating causal effects directly given a dataset requires that the data was obtained by a randomized experiment, an assumption likely to go unmet in most cases. In this paper, the authors propose a method to transform a causal estimate from a source dataset (meeting this assumption) to a target dataset. In this way, we can obtain causal estimates in related domains without the additional overhead of collecting new data.

**Questions For The Authors:**

A. Why was there no comparison to other methods? Please clarify if this is the first estimation protocol for causal effects when the target distribution is not a randomized experiment.
Besides using the density ratio as the importance weight (eqn 3), could other transformations be tested quickly? You could motivate better why this mapping was selected.

B. It is assumed that any crowdsourced dataset represents a randomized experiment and therefore does not have any confounding between the choice to read a text and the response to the text. Is it possible that other conditions of the crowdsourced experiment or motivations for participating in the experiment are confounding the response?

C. In the benchmark portion of the evaluation, the first (controlled setting) source and target datasets are obtained by splitting a single crowdsourced dataset into two parts which are expected to be distinct, and the average response is expected to differ between the two. How did you choose the partitions/categories and was there any pre-evaluation to check that the response was distinct?

Minor clarification questions:

D. An assumption of the target distribution $P^T$ is that it is absolutely continuous with respect to the source distribution $P^R$. Can you expand on any practical limitations this introduces? When would this not be satisfied?

**Reasons To Accept:**

* The results are convincing and sound; in three scenarios where the causal effect differs significantly between the source and target distributions, TEXT-TRANSPORT accurately estimates the effect in the target distribution indirectly through the source estimate.

* The authors do a great job motivating the problem, both conceptually and notationally. The Problem Setting is easy to read; equations 3 and 4 are intuitively explained. Overall the paper is a pleasure to read.

**Reasons To Reject:**

* There is no comparison with competing methods (if they exist)

**Reproducibility:**

4: Could mostly reproduce the results, but there may be some variation because of sample variance or minor variations in their interpretation of the protocol or method.

**Reviewer Confidence:**

3: Pretty sure, but there's a chance I missed something. Although I have a good feel for this area in general, I did not carefully check the paper's details, e.g., the math, experimental design, or novelty.

**Typos Grammar Style And Presentation Improvements:**

Some minor suggestions:

A. Intro lines 051-053: give an example scenario of confounding

B. At line 061 it is stated "effects estimated from randomized experiments may not generalize outside of the specific data on which they were conducted". Although this seems like an intuitive statement, it would be helpful to see it formalized if possible. If citations or a small sketch in the appendix could be added, it would help interested readers understand the problem better from the offset.

C. Organization: Section 5.3 would have been easier to read closer to sections 4.1-4.2, on the topic of the dataset distributions.

D. Notation: is $\tau^*$ (eqn 2) conventional notation for the natural effect? Should the $T$ at line 144 (and after) be a superscript?

---

> ### Author Rebuttal · Authors · 2023-08-28
>
> Thank you for your detailed feedback and for pointing out many strengths of our approach! We address your questions below.
>
> [**Question A, comparison to other methods**] As we mention in our Related Works section, a small number of prior studies have proposed estimators of some type of text causal effect from observational (i.e., non-randomized) data (Pryzant et al. 2018, Pryzant et al. 2021). However, the effects obtained from these methods are not directly comparable to those obtained using Text-Transport. The 2018 Pryzant et al. paper estimates word-level effects rather than linguistic attribute-level effects, limiting their interpretability. The 2021 Pryzant et al. paper estimates isolated effects rather than natural effects; but isolated effects are difficult to interpret in a natural language setting, as we highlight in lines 132-143 of our paper. Moreover, for both existing methods, the estimates of the causal effects are only valid if all potential confounders are measured in the data. This requirement is seldom fulfilled for text data, as (1) reader attributes are rarely recorded for the text datasets used in modern NLP, and (2) there are many varied factors that can influence both a person’s decision to read a text and the person’s response to the text (e.g., political affiliation, age, mood).
>
> [**Question A, motivation of density ratio**] Other transformations of the source distribution to the target distribution are indeed possible—for instance, the naive baseline described by reviewer aAKu, in which a predictive model trained on the source distribution is used to produce pseudo-labels on the target distribution, which are then averaged. The results of this baseline are shown below:
>
> | Dataset 	| Method | Distribution | Response | Lower CI | Upper CI |
> |-------------|:------:|:------------:|:--------:|:--------:|:--------:|
> | Amazon  	|	-   |	source	| 1.564	| 1.438	| 1.684	|
> | Amazon  	|	-   |	target	| 1.191	| 1.115	| 1.265	|
> | Amazon  	|   clf  |  transported | 1.243	| 1.149	| 1.322	|
> | Amazon  	|   LM   |  transported | 1.214	| 1.066	| 1.371	|
> | Amazon  	|  naive |  transported | 1.353	| 1.141	| 1.594	|
> | EmoBank 	|	-   |	source	| 3.229	| 3.217	| 3.242	|
> | EmoBank 	|	-   |	target	| 2.689	| 2.677	| 2.702	|
> | EmoBank 	|   clf  |  transported | 2.844	| 2.821	| 2.868	|
> | EmoBank 	|   LM   |  transported | 2.677	| 2.470	| 2.88 	|
> | EmoBank 	|  naive |  transported | 3.135	| 3.11 	| 3.161	|
> | Hate Speech |	-   |	source	| 0.135	| 0.117	| 0.152	|
> | Hate Speech |	-   |	target	| 0.263	| 0.244	| 0.283	|
> | Hate Speech |   clf  |  transported | 0.179	| 0.134	| 0.227	|
> | Hate Speech |   LM   |  transported | 0.246	| 0.000	| 1.000	|
> | Hate Speech |  naive |  transported | 0.138	| 0.080	| 0.200	|
>
> We can see that this type of transformation is empirically less effective than the density ratio transformation used in Text-Transport, as the naive estimator produces effect estimates that are farther from the true target effects. In fact, for the EmoBank and Hate Speech datasets, there is essentially no movement of the transported estimate toward the target distribution, while for the Amazon dataset, there is less movement compared to either implementation of the Text-Transport approach. Density ratios have been shown in the domain adaptation literature to be effective in helping models learn under distribution shift (see citations in our Related Works, lines 560-568). We will motivate the use of the density ratio in the context of the domain adaptation literature earlier in our paper, when we first introduce the method; and we will include the results of the naive baseline.
>
> [**Question B**] Yes, it is possible in the sense that certain types of people may be attracted to crowdworking platforms like Mechanical Turk or Prolific. However, this type of confounding (selection bias) can be present in any type of randomized experiment—that is, respondents in traditional randomized experiments or trials may also have specific motivations (often financial) for participating that make them different from the overall population. Tackling this issue has been the focus of a large literature, using techniques related to the method we develop here. For example, the work of Egami and Hartman (2021), Pearl and Bareinboim (2022), and Tipton (2014) that we discuss in the Related Work section approach this issue as either a covariate- or label-shift problem. In principle, it is possible to combine our method with the techniques in this literature to generalize to a target domain specified by both the text corpus and the distribution of respondents. We hope to examine this in future work.
>
> [**Question C**] For the Amazon and Hate Speech datasets, we chose partitions that already existed in the datasets—for instance, the Amazon dataset is divided into product categories, and the Hate Speech dataset contains posts from both Reddit and Gab, making for an intuitive split. For the EmoBank dataset, as we mention in Section 4.2, we chose to split along writer-intended valence to create a partially controlled data setting in which the source and target differ on a single attribute that is known to us but hidden from the models. We did check that the average responses were distinct in the source and target distributions; the separation of responses across source and target can be seen in Figure 3 of our paper.
>
> [**Question D**] The absolute continuity assumption is violated if a text that would *never* occur in the source distribution could possibly occur in the target distribution. Therefore, this assumption may not be satisfied if the source and target distributions are completely unrelated and non-overlapping, even in latent space. Practically speaking, this means that it may not be possible to transport effects between distributions that are extremely different: for instance, from a corpus of technical manuals to a corpus of Shakespearean poetry.
>
> [**Minor suggestions**] Thank you for these helpful suggestions! We will implement them in the next version of the paper. Following suggestion B, we will include citations for this statement (much of the literature on transportability is premised on this notion, e.g. the transportability papers cited in our Related Works: Tipton 2014, Barenboim and Pearl 2021). Regarding suggestion D, $\tau^*$ is conventional notation for a general causal effect. We agree that superscripting $T$ may be clearer notationally and will make this change.

---

### Meta-Review · Area_Chair_2uqh · 2023-09-18

**Recommendation:** 5

**Metareview:**

This paper proposes text-transport, a method to transform a causal estimate from a source to a target dataset. With this method, one can obtain causal estimates in related domains without the additional overhead of collecting new data. Two methods are proposed: one based on a binary classifier to distinguish between source and target domains and one based on prompting an LLM. The authors empirically show their framework's validity on three datasets and provide additional insights for the use case of hate speech detection. Comprehensive experiments and a thorough analysis of results support the paper's claims. The paper is well-written and exciting to read.

The reviewers have raised concerns about comparing with baseline methods and the rationale behind the proposed method. These concerns have been addressed by the authors in their rebuttals, which need to be incorporated into the final version of the paper.

---

### Decision · Program_Chairs · 2023-10-07

**Decision:**

Accept-Main

**Comment:**

This paper proposes text-transport, a method to transform a causal estimate from a source to a target dataset. With this method, one can obtain causal estimates in related domains without the additional overhead of collecting new data. Two methods are proposed: one based on a binary classifier to distinguish between source and target domains and one based on prompting an LLM. The authors empirically show their framework's validity on three datasets and provide additional insights for the use case of hate speech detection. Comprehensive experiments and a thorough analysis of results support the paper's claims. The paper is well-written and exciting to read.

The reviewers have raised concerns about comparing with baseline methods and the rationale behind the proposed method. These concerns have been addressed by the authors in their rebuttals, which need to be incorporated into the final version of the paper.